# Macroelement and Microelement Levels in the Urine in Experimental Acanthamoebiasis

**DOI:** 10.3390/pathogens12081039

**Published:** 2023-08-14

**Authors:** Natalia Łanocha-Arendarczyk, Karolina Kot, Irena Baranowska-Bosiacka, Patrycja Kupnicka, Dagmara Przydalska, Aleksandra Łanocha, Dariusz Chlubek, Iwona Wojciechowska-Koszko, Danuta Izabela Kosik-Bogacka

**Affiliations:** 1Department of Biology and Medical Parasitology, Pomeranian Medical University in Szczecin, 70-204 Szczecin, Poland; natalia.lanocha.arendarczyk@pum.edu.pl (N.Ł.-A.); karolina.kot@pum.edu.pl (K.K.); 30555@student.pum.edu.pl (D.P.); 2Department of Biochemistry and Medical Chemistry, Pomeranian Medical University in Szczecin, 70-204 Szczecin, Poland; irena.baranowska.bosiacka@pum.edu.pl (I.B.-B.); patrycja.kupnicka@pum.edu.pl (P.K.); dariusz.chlubek@pum.edu.pl (D.C.); 3Department of Haematology and Transplantology, Pomeranian Medical University in Szczecin, 70-204 Szczecin, Poland; aleksandra.lanocha@pum.edu.pl; 4Department of Microbiology, Immunology and Laboratory Medicine, Pomeranian Medical University, 70-111 Szczecin, Poland; iwona.wojciechowska.koszko@pum.edu.pl; 5Independent Laboratory of Pharmaceutical Botany, Pomeranian Medical University in Szczecin, 70-204 Szczecin, Poland

**Keywords:** macroelements, microelements, urine, *Acanthamoeba* sp.

## Abstract

Free-living amoebas can impact the excretion of macroelements and microelements in urine. The aim of the present study was to examine the concentrations of macroelements, including calcium (Ca), phosphorus (P), sodium (Na), potassium (K), and magnesium (Mg), as well as microelements such as manganese (Mn), zinc (Zn), copper (Cu), iron (Fe), and chromium (Cr), in the urine during acanthamoebiasis while considering the host’s immunological status. This is the first study to show an increase in urinary excretion of Ca, Mn, Cu, Fe, Na, and Cr, along with a decreased excretion of K, in immunocompetent mice 16 days post *Acanthamoeba* sp. infection. In the final phase of infection (24 dpi), there was a further decrease in urinary K excretion and a lower level of P in *Acanthamoeba* sp. infected immunocompetent hosts. During acanthamoebiasis in immunosuppressed hosts, increased excretion of Zn, Fe, and Cr was observed at the beginning of the infection, and increased Na excretion only at 16 days post *Acanthamoeba* sp. infection. Additionally, host immunosuppression affected the concentration of Fe, Cr, Zn, Cu, Mn, and Ca in urine.

## 1. Introduction

Opportunistic infection by *Acanthamoeba* sp., a pathogenic free-living amoeba, primarily targets the central nervous system and typically occurs in patients undergoing intensive steroid therapy, organ transplant recipients, HIV-infected individuals, and children [1,2]. *Acanthamoeba* sp. spreads to different organs, including the kidneys, from its initial location in the skin, subcutaneous tissue, or lungs [3,4]. The brain *Acanthamoeba* infection is often accompanied by vague symptoms such as headache, nausea, irritability, dizziness, and a low-grade fever. Additionally, it can be misdiagnosed with bacterial leptomeningitis, tuberculous meningitis, or viral meningitis [5]. Little is known about the complex pathomechanisms that occur in the host’s body during a multi-organ invasion by *Acanthamoeba*. Given the high mortality and swift disease progression, comprehending the processes and fundamental mechanisms within the parasite-host system presents an opportunity to curtail the invasion.

Currently, there is no definitive and effective therapy for free-living ameba infections, underscoring the critical importance of the immune system’s response. The survival of these pathogens and the activation of the host’s immune system depend on the available trace elements, which are included in coenzymes that activate immune cells and may influence the production, maturation, and activity of leukocytes [6]. However, little is known about the status of macronutrients and micronutrients excreted in the urine during parasitic infections, even though the kidneys are key detoxification organs and urine is a convenient material for rapid analysis. In rabbits infected with *Trypanosoma brucei*, increased excretion of magnesium (Mg) and zinc (Zn) in the urine has been observed [7]. Furthermore, children with schistosomiasis show significantly higher levels of iron (Fe) and copper (Cu) in urine, while manganese (Mn) and cadmium (Cd) concentrations are reduced in patients with acute *Schistosoma haematobium* infection with hematuria [8]. Łanocha-Arendraczyk et al. [9] found higher nephric selenium (Se) concentrations in the immunocompetent mice compared to the control host.

The loss of Fe, molybdenum (Mo), Cu, and Zn has been linked to a higher burden of gastrointestinal nematodes due to excessive intakes of Mo, Fe, and Cu [10]. Imbalances in trace element metabolism can lead to metabolic disorders and pathophysiological processes [11]. Macronutrient and micronutrient deficiencies often coexist with infectious diseases, including opportunistic parasite infections, resulting in complex interactions that amplify harmful clinical effects [12,13]. These relationships are common in individuals with weakened immune systems [14]. Some trace elements, such as Zn, Fe, and Se, can modulate immune function, influence host susceptibility to infection, and protect cells from oxidative stress [15].

Changes in the concentrations of elements in the urine may be a result of the inflammatory process, disturbances in hormonal balance, absorption, and excretion, and the effects of medications, including immunosuppressive drugs. Hence, they provide useful information about the state of the organism’s health [16,17]. In the scientific literature, no information has been found regarding the concentration of macroelements and microelements in relation to *Acanthamoeba* infection. Furthermore, the mechanisms underlying the increase or decrease in element levels in the urine of *Acanthamoeba* infected hosts are not understood. Acanthamoebiasis usually occurs in immunocompromised or debilitated patients. However, despite the immunocompromised state being a risk factor, there are reports of severe disease in immunocompetent patients [3,4,5]. Therefore, in the case of *Acanthamoeba* sp. infection, it is crucial to understand the course of infection in hosts with varying immunological status. Thus, to clarify and elucidate the status of macro- and microelements during acanthamoebiasis in relation to the host immune status, the purpose of this study was to examine the concentrations of macroelements including calcium (Ca), phosphorus (P), sodium (Na), potassium (K), magnesium (Mg), and microelements such as manganese (Mn), zinc (Zn), copper (Cu), iron (Fe), and chromium (Cr) in the urine during acanthamoebiasis, while taking into account the host’s immunological status.

## 2. Materials and Methods

### 2.1. Design of the Experimental Model

The experimental design for acanthamoebiasis (an animal infection study involving immunosuppressed mice) has been extensively detailed in our prior publications [18,19]. The study comprised 96 male Balb/c mice, aged between 6 and 10 weeks, which were procured from a licensed breeder at the Centre of Experimental Medicine, Medical University of Białystok, Poland. All animal-based experimental protocols received approval from the Local Ethics Committees for Experiments on Animals in Szczecin (Approval No. 29/2015, dated 22 June 2015) and Poznań (Approval No. 64/2016, dated 9 September 2016).

The mice were allocated into four groups:Group C (*n* = 18): immunocompetent control group;Group A (*n* = 30): immunocompetent mice infected with *Acanthamoeba* sp.;Group AS (*n* = 30): immunosuppressed mice infected with *Acanthamoeba* sp.;Group CS (*n* = 18): immunosuppressed uninfected mice.

Immunosuppressed groups (AS and CS) were administered 0.22 mg (10 mg/kg) of methylprednisolone sodium succinate (MPS, Solu-Medrol, Pfizer, Europe MA EEIG, Bruxelles; Belgium) intraperitoneally (i.p.) five times (on days −4, −3, −2, −1, and 0) preceding the ameba inoculation. Groups A and AS received an intra-nasal inoculation of a 3 μL suspension housing 10–20 thousand ameba trophozoites. These amoebas (AM 22 strain) were isolated from the bronchoaspirate of a hemato-oncology patient suffering from acute septic shock [20]. The control groups (C and CS) were administered an equal volume of sterile physiological solution (3 μL of 0.9% NaCl solution).

The experiment ended on the 8th, 16th, and 24th days post-infection (dpi), after which the animals were euthanized using an overdose of pentobarbital sodium (Euthasol vet, FATRO, Bologna, Italy) (2 mL/kg body weight). During animal dissection, urine samples were collected via bladder injection, stored in sterile 1.5 mL Eppendorf tubes, and subsequently frozen at −20 °C until further chemical analysis. Urine samples were collected from infected animals that showed clinical signs of amoeba infection. A comprehensive outline of the experimental design and the timeline of the experiment can be found in Figure 1.

### 2.2. Analysis of Macro and Microelements

The urine samples were digested using 1 mL of 65% HNO_3_, 0.5 mL of 30% H_2_O_2_, and 1 mL of 35% HNO_3_ (SuprapurMerck^®^, Darmstadt, Germany). Before measurement, each digested urine sample was diluted in a 1:20 ratio using the following formula: 0.5 mL of digested sample, 1 mL of a 5 mg/L nitrate solution dissolved in HNO_3_, 1 mL of 1% Triton (Triton X-100, Sigma, Poznań, Poland), and 7.5 mL of 0.075% HNO_3_ (Suprapur, Merck, Darmstadt, Germany). The samples were kept at 4 °C until analysis. The concentrations of Ca, Mn, K, Zn, Cu, Fe, Na, Cr, P, and Mg were determined by spectrophotometric atomic absorption in an inductively coupled argon plasma (ICP OES) using an ICAP 7400 instrument (Thermo Scientific, Waltham, MA, USA). A multi-element standard solution in various dilutions (ICP multi-element standard solution IV, Merck, Darmstadt, Germany) was used to create the calibration curve. Deionized water (Direct Q UV, Millipore, ~18.0 MΩ) was used to prepare the solutions. Each sample was measured three times, and a standard curve was plotted before each analysis. The analysis of the sample was performed only if the correlation linearity of the standard curve (R^2^) was greater than 0.999. The instrumental detection limits (LoD) for ICP OES of the analyzed metals are presented in Table 1, and the wavelength used for each element, LoD, and limits of quantitation (LoQ) for each metal are given in Appendix A, Table A1. In this study, yttrium was used as an internal standard (Y = 1 mg/L Y(NO_3_)_3_ in HNO_3_ 0.5 mol/L; Merk).

The quality of the analytical process was verified by analyzing a certified reference material, bovine muscle (8414 NIST, Bovine Muscle Powder, National Institute of Standards and Technology, Gaithersburg, MD, USA). The metal concentrations of the reference materials are presented in Table 1. The concentrations of elements in the urine are expressed as mg/L.

### 2.3. Statistical Analysis

For the concentrations of each element in the urine, we calculated the median (Med), arithmetic mean (AM), ±standard deviation (SD), and range. Since the concentrations of elements frequently deviated from the expected normal distribution (as confirmed by the Shapiro-Wilk test), we used non-parametric tests for the comparisons (Kruskal-Wallis test, K-W; Mann-Whitney U test, M-W). Additionally, we calculated Spearman’s rank correlation (r_s_) coefficients and interpreted the correlation strengths based on Bryman and Cramer’s [21] classification: 0.8–1 as very strong, 0.6–0.79 as strong, 0.4–0.59 as moderate, 0.2–0.39 as weak, and 0–0.19 as very weak. All calculations were performed using Statistica v 6.0 software, and statistical significance was determined at *p* < 0.05.

## 3. Results

The concentrations of the studied macroelements and microelements in the urine of *Acanthamoeba* sp. infected mice, categorized by their immunological status, are presented in Table 2 and Table 3. In the examined urine samples from the A and AS groups, the greatest concentrations were observed for Na and K and the lowest for Cr and Cu.

Significant differences were observed in the concentrations of Ca, Mn, Zn, Na, and Cr in the urine of immunocompetent *Acanthamoeba* sp. infected mice on different days post-infection. In the urine of immunocompetent mice, a decrease in Ca, Mn, Zn, Na, and Cr was found at 8 days post-Acanthamoeba sp. infection, followed by an increase at 16 dpi and a re-decrease at 24 dpi (H = 12.11, *p* < 0.02; H = 10.80, *p* < 0.04; H = 11.73, *p* < 0.03; H = 7.07, *p* < 0.03; H = 10.56, *p* < 0.01, respectively) (Table 2).

No significant differences were observed in immunocompromised *Acanthamoeba* sp. infected mice on different days post-infection. There were no statistically significant differences between the groups of infected and uninfected immunocompetent mice on the eighth day post Acanthamoeba sp. infection. At 16 dpi, we found a statistically significant lower K concentration (U = 2.0, *p* < 0.02) in the urine of immunocompetent *Acanthamoeba* sp. infected mice than in the uninfected group. Additionally, we found statistically significant higher levels of Ca (U = 0, *p* < 0.01), Mn (U = 0, *p* < 0.01), Zn (U = 0, *p* < 0.01), Cu (U = 0, *p* < 0.01), Fe (U = 0, *p* < 0.01), Na (U = 1.0, *p* < 0.01), and Cr (U = 0, *p* < 0.01) in the urine of immunocompetent infected mice than in the control group at 16 dpi (Table 2). At 24 dpi, significantly lower concentrations of K and P were observed in the urine of immunocompetent *Acanthamoeba* sp. infected mice than in the uninfected group (U = 3.0, *p* < 0.03; U = 0, *p* < 0.01, respectively).

There was a statistically significant higher level of Zn, Fe, and Cr in the urine of immunocompromised *Acanthamoeba* sp. infected mice at 8 dpi (U = 6.0, *p* < 0.05; U = 5.0, *p* < 0.03; U = 6.0, *p* < 0.05, respectively) and a significantly higher Na level in *Acanthamoeba* sp. infected mice at 16 dpi (U = 2.0, *p* < 0.04) compared to the uninfected immunocompromised mice. There were no statistically significant differences between the groups of infected and uninfected immunocompromised mice at 24 dpi.

The concentrations of Mn, Fe, and Cr in the urine at the beginning of infection were found to be influenced by the host’s immunological status. The concentration of Mn (U = 5.0, *p* < 0.03), Fe (U = 3.0, *p* < 0.02), and Cr (U = 1.0, *p* < 0.01) in the urine of immunosuppressed infected mice was higher at 8 dpi than in the immunocompetent Acanthamoeba sp. infected mice. At 16 dpi, the concentrations of Ca (U = 0, *p* < 0.01), Mn (U = 0, *p* < 0.01), Zn (U = 0, *p* < 0.01), Cu (U = 0, *p* < 0.01), and Fe (U = 0, *p* < 0.01) decreased in the immunosuppressed mice compared with the immunocompetent mice, while at 24 dpi, K levels increased in the Acanthamoeba sp. infected immunosuppressed hosts (U = 2.0, *p* < 0.04) compared with the immunocompetent mice. The statistical analysis focused on examining the relationships between macro- and micro-elements in the urine of immunocompetent (A) and immunosuppressed (AS) mice infected with *Acanthamoeba* sp.

At 8 dpi in the Acanthamoeba sp. infected immunocompetent host, very strong positive correlations (r_s_ = 0.90) were observed between Mg and K, Mg and Na, Mg and P, and Mg and Cr concentrations in the urine. At 16 dpi, the strongest relationships (r_s_ = 0.94) were noted between Ca and Mn, Ca and P, P and Cu, and P and Fe. Furthermore, statistically significant positive correlations were found between Ca and Cu (r_s_ = 0.83), Zn and Fe (r_s_ = 0.89), Cu and Fe (r_s_ = 0.89), Zn and Cu (r_s_ = 0.83), P and Mn (r_s_ = 0.83), and P and K (r_s_ = 0.83), while at 24 dpi, only two positive correlations were observed, between Ca and Cu (r_s_ = 0.94) and Cr and Na (r_s_ = 0.89). All relationships were synergistic. At 8 dpi in the Acanthamoeba sp. infected immunosuppressed group, very strong, statistically significant positive correlations (r_s_ = 0.98) were detected between Fe and Mn, Fe and Zn, Fe and Cu, and Cu and Cr. Additionally, significant positive correlations were observed between Zn and Mn (r_s_ = 0.96), Cu and Zn (r_s_ = 0.95), Cr and Fe (r_s_ = 0.95), Ca and Mn (r_s_ = 0.93), Ca and Cu (r_s_ = 0.93), Ca and Fe (r_s_ = 0.90), Cr and Zn (r_s_ = 0.90), Ca and Cr (r_s_ = 0.86), Mn and Cr (r_s_ = 0.88), Ca and Zn (r_s_ = 0.88), and P and Cr (r_s_ = 0.71).

At 16 dpi, in the AS group, strong positive correlations (r_s_ = 0.90) were noted between Mg and K, Mg and Na, and Mg and Cr. Additionally, relationships were detected between Ca and Mn, Cr and Mn, Zn and K, Zn and Na, and Fe and Cr.

At 24 dpi, positive relationships (r_s_ = 0.90) were observed in mice with reduced immune function between Zn and Ca, Zn and Mn, Zn and Cu, Zn and Cr, Cu and K, Fe and K, Fe and Cu, Na and K, and Mg and K.

The statistical analysis did not reveal any significant relationships between macro- and microelements in the urine of immunocompetent and immunosuppressed mice infected with *Acanthamoeba* sp., Kidney Injury Molecule-1 (KIM-1), and neutrophil gelatinase- associated lipocalin (NGAL).

## 4. Discussion

Inflammation in critical detoxification organs such as the kidneys can lead to alterations in blood biochemical indicators, including key renal function biomarkers such as urea and creatinine [22]. Our previous study, employing the same experimental model of acanthamoebiasis, revealed no notable differences in the serum concentrations of creatinine and urea [9], despite observing changes in inflammatory parameters within the kidneys [23]. In our earlier research, we noted that amoebas traveling through the bloodstream infiltrate the kidneys and can induce their dysfunction, potentially affecting the excretion of elements in the urine [23].

There are studies indicating that urinary trace element analysis can serve as an indicator of bacterial or viral infections as well as disease severity [23,24,25,26,27,28,29]. However, the connections between urinary trace element levels and parasitic infections, including acanthamoebiasis, remain relatively unexplored. Most research concerning element analysis during parasitic infection has focused on changes in trace elements within the serum but rarely in the urine. This study is the first to show that free-living amoebas can impact the excretion of macroelements and microelements in urine.

### 4.1. Macroelements in the Urine

The kidneys play a central role in maintaining calcium homeostasis by finely regulating renal Ca excretion. Foley and Boccuzzi [30] noted that Ca is required for muscle contraction and nerve signaling. Calcium also plays a role in immune and regenerative processes, influencing macrophage motility and antibody synthesis [31]. It has been shown that Ca has an important role in the signaling of different cellular processes in parasitic infections such as amoebiasis, cryptosporidiosis, and toxoplasmosis, including development and pathogenesis [32,33]. Increased urinary excretion of Ca may occur in kidney diseases and progressive inflammation. In this study, we found significantly higher levels of Ca in the urine at 16 dpi in immunocompetent *Acanthamoeba* sp. infected mice compared to the control group. At the same time of infection, we also noted increased urinary excretion of Ca in immunocompetent *Acanthamoeba* sp. infected hosts compared to the group of immunosuppressed infected mice.

Sodium plays a crucial role in nerve cell excitation, and elevated Na levels in urine could result from increased renal loss of this element. In the course of acanthamoebiasis, we found significantly increased urinary excretion of Na only at 16 dpi in immunocompetent and immunosuppressed hosts infected with *Acanthamoeba* sp. compared to the control groups. Phosphorus has a physiological role in the body and is involved in skeletal mineralization, nerve impulse transmission, blood clotting, and metabolic pathways [34]. It is a human intracellular anion that participates in maintaining the acid–base balance in the body by creating buffer systems in the urine [35]. At the last stage of acanthamoebiasis, we noted a decreased P concentration in the urine of immunocompetent hosts compared to the control group.

Potassium is mainly excreted through the urine, and the kidneys play an important role in its excretion. In cases of acute and/or chronic renal failure, when glomerular filtration is reduced to 10–15%, the amount of filtered K is not sufficient to maintain a normal balance [36]. The excretion of K in urine depends not only on the size of glomerular filtration but also on the active role of renal tubules [37]. The redistribution of K in the body depends on the levels of insulin, catecholamines, and aldosterone [38]. In an *Acanthamoeba* sp. infected immunocompetent host, a significant decrease in K urine concentration was observed at 16 and 24 days post *Acanthamoeba* sp. infection compared to the control group, which may result from increased K loss through the gastrointestinal tract (diarrhea, vomiting) [39].

The kidneys play an important role in regulating the normal level of magnesium in the body, and the excretion of Mg by them accounts for approximately 5%, with the remaining amount undergoing reabsorption mainly in the ascending limb of the Henle’s loop in the renal tubules [40]. It has been shown that Mg deficiency leads to increased excretion of K in urine, and some authors have demonstrated a close relationship between Mg and the inflammatory response [41]. Chronic Mg deficiency may result in increased oxidative stress and low-grade inflammation [42]. In the final phase of *Acanthamoeba* sp. infection (24 dpi), a decrease in Mg levels in urine (but not significant) was observed regardless of the host’s immunological status compared to the control group. A similar relationship was observed in the urine of patients with renal cell carcinoma [29,43].

### 4.2. Microelements in the Urine

Manganese in urine represents the excretion of its excess from the body. This element has a special affinity for mitochondrial-rich tissues such as the liver [44]. Hepatic dysfunction may result in Mn accumulation, which could lead to higher urinary Mn levels in acanthamoebiasis in immunocompetent hosts at 16 dpi. In our previous study, we found upregulation of serum aspartate aminotransferase and alanine aminotransferase (ALT) in immunocompetent *Acanthamoeba* infected mice, which may indicate liver dysfunction during amoebic infection [9]. Some researchers have found that diabetics with liver disorders tend to have significantly increased excretion of Mn in their urine [45].

During infection and inflammation, zinc levels decrease [46]. Diminished levels of essential elements increase susceptibility to infection, oxidative stress, and proinflammatory processes [47,48]. Zinc is critical for the survival and virulence of pathogenic microorganisms, and homeostasis disorders affect both the innate and adaptive immune systems [49]. In this study, increased urinary excretion of Zn was observed at 8 and 16 days post *Acanthamoeba* sp. infection in immunosuppressed and immunocompetent hosts, respectively. Low Zn is associated with impaired immune function and a poor prognosis in sepsis, malaria, and HIV infection [50]. Kahvaz et al. [51] observed lower levels of Fe and Zn in patients’ serum with cutaneous leishmaniasis (CL); however, Cu levels were higher. The authors suggested that the status of trace element serum levels in CL patients probably depends on IL-1 and TNF-alpha cytokines secreted by activated macrophages as part of the response to parasite infection [52].

Copper levels in urine are typically elevated in urinary tract infections, and Cu plays a role in guarding against this illness [53]. Copper possesses antiparasitic qualities and positively impacts host immunity, playing a crucial role in immune function and antioxidant defense [54]. Arinola [8] discovered notably elevated Cu levels in the urine of Nigerian children infected with *Schistosoma haematobium*. Similarly, we observed an increased Cu concentration in immunocompetent hosts compared to the control group. The increased excretion of Zn and Cu (trace elements with protective biological properties) during acanthamoebiasis in immunocompetent hosts could result in reduced levels of these elements in the body. This process might be linked to heightened mobilization to counter oxidative events caused by or arising from inflammatory processes. Diminished levels of Zn and Cu in *Acanthamoeba*-infected hosts might contribute to susceptibility to oxidative stress and pro-inflammatory processes (as confirmed in our prior research on the lungs, brain, and kidneys).

Iron is crucial for regulating immune cell proliferation and function. Chronic parasitic infections, such as malaria, schistosomiasis, and hookworm infection, are major global causes of Fe deficiency [55]. In the case of *Schistosoma* sp. infection, Fe deficiency anemia is associated with blood loss in feces and urine. During acanthamoebiasis, we noted increased urinary excretion of Fe in immunosuppressed and immunocompetent mice at 8 and 16 days post *Acanthamoeba* sp. infection. Some authors suggest that increased urinary Fe concentrations have been associated with renal injury. Raaij et al. [56] demonstrate that increased circulating Fe levels and subsequent filtration, as well as insufficient tubular reabsorption, are associated with increased urinary Fe concentrations and, importantly, renal injury.

The function of chromium during inflammation remains unclear. It affects various components of the immune system and may result in immunostimulation or immunosuppression [57]. Increased urinary excretion of Cr has been observed in patients with hypertension and impaired kidney function [58]. Moreover, Cr may improve glucose tolerance through insulin resistance [59,60]. It is possible that disturbances in glucose homeostasis during progressing inflammatory processes are related to higher levels of urinary Cr loss. Some authors have observed that hyperglycemia is a poor prognostic factor in severe infections such as malaria and COVID-19 [61]. Similarly, Zeng et al. [28] found that disturbances in urinary concentrations of Cr and Mn are associated with severe illness, which was also observed in mice during *Acanthamoeba* sp. infection.

### 4.3. Immunosuppression and Elements in the Urine

Disturbances in trace element metabolism can be caused, among other things, by immunosuppressive drugs such as cyclosporin A and tacrolimus, which are used to prevent the rejection of vascularized organ transplants [62]. The drugs inhibit the proliferation of T and B cells and can induce inhibition of reabsorption in the kidneys and increased excretion of trace elements in urine [63]. In addition, it has been shown that immunosuppressive drugs, which increase the concentration of Cu and Zn in the blood serum, also reduce the concentration of Na [64]. Wilk et al. [61] found higher levels of Fe in patients treated with immunosuppressive drugs, including mycophenolate mofetil (MMF).

In this study, we noted that immunosuppression affected the concentration of some elements. In immunosuppressed infected mice after administration of methylprednisolone (a synthetic glucocorticoid with potent and long-acting anti-inflammatory, antiallergic, and immunosuppressive effects), we found higher Mn, Fe, and Cr levels compared to the infected immunocompetent group at the beginning of *Acanthamoeba* sp. infection. The reverse trend was observed at 16 days post *Acanthamoeba* sp. infection, where we found decreased excretion of Ca, Mn, Zn, Cu, and Fe in the urine of mice with reduced immunity compared to immunocompetent hosts. Some researchers have observed that increased loss of K may be associated with the use of corticosteroids [65], which was confirmed in this study in infected mice subjected to steroid-induced immunosuppression at 24 days post *Acanthamoeba* sp. infection.

Increased excretion of trace elements in urine during the disease can lead to disruption of mineral balance in the body, including deficits in certain elements, and potentially heighten an individual’s susceptibility to the disease [66]. Some authors have proposed that during the inflammatory process, especially in immunosuppressed patients such as those with HIV, supplementing trace elements through therapeutic interventions is beneficial [28]. It has been demonstrated that chromium, iron, selenium, and zinc can impact HIV progression and support its treatment. By monitoring the excretion of trace elements in urine during acanthamoebiasis, it may be possible to prevent deficiencies through appropriate supplementation, similar to other diseases, which could influence the process and pace of treatment. However, achieving this objective requires further research.

This work has certain limitations. Due to the early and humane endpoint of the experiment, which was associated with the strong symptoms of the disease, the number of animals in the groups was considerably reduced. Therefore, in future studies, it is necessary to increase the number of animals infected with *Acanthamoeba* sp. Secondly, the concentration of trace elements was only examined in urine. However, to track the impact of *Acanthamoeba* sp. on the host’s water-electrolyte balance, the concentration of elements in the blood, kidneys, and liver should be analyzed. Furthermore, it would be interesting to investigate the hormonal balance, especially by examining the levels of parathyroid hormone and insulin and estimating glucose homeostasis. Additionally, the analysis of elements could be expanded to include selenium, which is essential for an adequate immune response and possesses anti-inflammatory and antioxidant properties. The method of collecting urine samples via bladder injection can also be considered a drawback of this study. Using metabolic cages for the collection of 24-h urine samples would be a more suitable approach for future research.

## 5. Conclusions

The results of this study demonstrate that free-living amoebas can affect the excretion of macro and microelements in urine, potentially paving the way for further investigation into element metabolism. This study is the first to demonstrate that the 16th day of *Acanthamoeba* sp. infection is crucial for the excretion of Ca, Mn, Cu, Fe, Na, Cr, and K in immunocompetent mice.

In immunosuppressed hosts during acanthamoebiasis, there was an initial increase in Zn, Fe, and Cr excretion, along with a later increase in Na excretion on the 16 day post *Acanthamoeba* sp. infection. Additionally, host immunosuppression influenced the concentration of Fe, Cr, Zn, Cu, Mn, and Ca in urine. The observation of changes in urinary element concentrations in *Acanthamoeba* sp. infected hosts may expand our knowledge regarding the pathophysiology of disseminated acanthamoebiasis.

## Figures and Tables

**Figure 1 pathogens-12-01039-f001:**
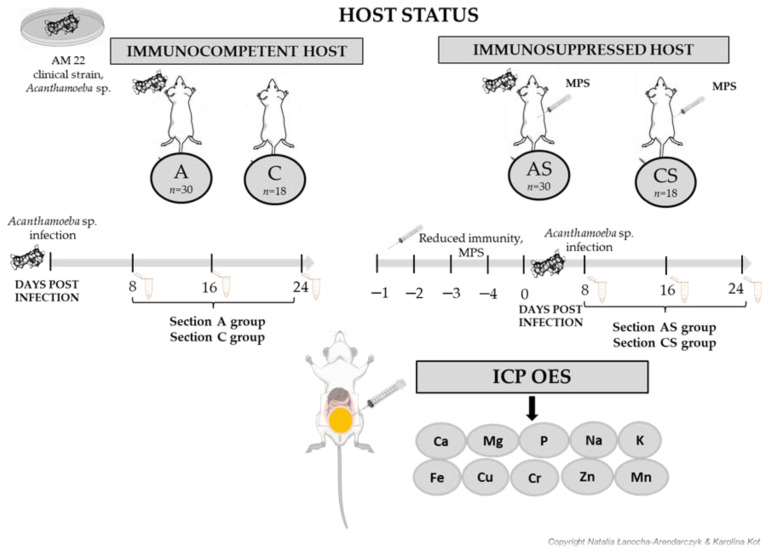
Schematic illustration of an experimental protocol to evaluate the levels of macroelements and microelements in the host urine infected with *Acanthamoeba* sp. The study utilized male Balb/c mice aged between 6 and 10 weeks at the onset of the experiment. The mice were divided into two categories based on their immunological status. The mice assigned to the AS and CS groups were subjected to an immunosuppression regimen involving methylprednisolone sodium succinate (MPS, Solumedrol) for four days before the *Acanthamoeba* sp. infection. Subsequently, the A and AS groups were infected with *Acanthamoeba* sp. trophozoites (AM 22 strain). Following this, at 8, 16, and 24 days post-infection (dpi), the mice were euthanized. Urine samples were then collected directly from the bladder for chemical analysis. To evaluate the levels of macroelements and microelements in the urine, the spectrophotometric atomic absorption method in an inductively coupled argon plasma (ICP OES) was employed. (A—*Acanthamoeba* sp. infected immunocompetent mice (*n* = 30); AS—immunosuppressed mice infected with *Acanthamoeba* sp. (*n* = 30); C—immunocompetent control group (*n* = 18); CS—immunosuppressed uninfected mice (*n* = 18)).

**Table 1 pathogens-12-01039-t001:** Analytical results for the content of the studied elements in the certified reference material, Bovine Muscle NIST-SRM 8414 (*n* = 3), along with the detection limits of the elements. The reference values (RV) and observed values (OV) are given in mg/L as the arithmetic mean ± standard deviation. The percentage expresses the OV/RV ratio.

Element	Detection Limit (LoD; mg/L)	Elements Concentrations in Bovine Muscle RM 8414 (mg/L), *n* = 3
Ca	0.0188	RV: 145.00 ± 20.00
		OV: 156.23 (107.7%)
P	0.0027	RV: 8360 ± 450
		OV: 8213.7 (98.3%)
Na	0.0205	RV: 2100 ± 80
		OV: 1789.2 (85.2%)
K	0.0574	RV: 15170 ± 370
		OV: 12,185.8 (80.3%)
Mg	0.0005	RV: 960 ± 95
		OV: 897.7 (93%)
Mn	0.0001	RV: 0.37 ± 0.09
		OV: 0.477 (129.1%)
Zn	0.0001	RV: 142 ± 20
		OV: 150.28 (195.8%)
Cu	0.0006	RV: 2.84 ± 0.45
		OV: 3.02 (106.5%)
Fe	0.0007	RV: 71.2 ± 9.2
		OV: 66.7 (93.7%)
Cr	0.0001	RV: 0.071 ± 0.0038
		OV: 0.204 (287.9%)

**Table 2 pathogens-12-01039-t002:** Macroelement and microelement concentrations (mg/L) in the urine of immunocompetent mice 8, 16, and 24 days post *Acanthamoeba* sp. infection (dpi) compared to the control group (*A*, immunocompetent *Acanthamoeba* sp. infected mice; C, immunocompetent uninfected mice (control group); AM arithmetic mean; SD standard deviation; *p* level of significance; * *p* < 0.01, ** *p* < 0.02, *** *p* < 0.03 for the significance of the difference vs. control (Mann-Whitney U test).

Element	Parameter	A			C		
		8	16	24	8	16	24
Ca	AM ± SD	79.8 ± 35.9	567.7 ± 410.6 *	31.6 ± 24.1	42.7 ± 30.7	88.5 ± 71.8 *	51.9 ± 57.3
	Med	76.8	567.7	24.1	32.8	52.5	51.9
P	AM ± SD	196.0 ± 103.8	305.47 ± 48.41	209.46 ± 137.90 *	161.7 ± 105.3	177.1 ± 96.1	2458.6 ± 1120.9 *
	Med	161.6	305.47	171.29	161.7	172.2	2458.6
Na	AM ± SD	2158.1 ± 1573.3	4587.8 ± 3771.7 *	1087.8 ± 586.2	1941.3 ± 121.9	1507.6 ± 388.5 *	1105.0 ± 85.9
	Med	1201.9	4587.8	1251.6	1941.3	1416.5	1105.0
K	AM ± SD	6846.8 ± 3621.3	6672.3 ± 569.2 **	4775.8 ± 1961.9 ***	6846.5 ± 1536.3	8873.7 ± 1949.3 **	8872.5 ± 3233.7 ***
	Med	5127.3	6672.3	4295.5	6846.5	8873.7	8872.1
Mg	AM ± SD	394.2 ± 199.9	365.2 ± 90.6	254.7 ± 79.5	271.6 ± 97.3	292.3 ± 76.9	431.5 ± 200.5
	Med	263.7	365.2	250.7	271.6	308.7	431.5
Mn	AM ± SD	0.02 ± 0.003	0.49 ± 0.39 *	0.03 ± 0.03	0.03 ± 0.01	0.03 ± 0.02 *	0.04 ± 0.02
	Med	0.02	0.49	0.02	0.03	0.03	0.04
Zn	AM ± SD	4.50 ± 2.0	34.66 ± 29.25 *	2.28 ± 2.15	3.0 ± 0.43	2.84 ± 1.13 *	6.44 ± 3.89
	Med	4.47	34.66	1.37	3	2.84	6.44
Cu	AM ± SD	0.22 ± 0.08	3.00 ± 2.91 *	0.17 ± 0.11	0.27 ± 0.04	0.22 ± 0.09 *	0.23 ± 0.11
	Med	0.23	3.00	0.15	0.27	0.21	0.23
Fe	AM ± SD	0.35 ± 0.11	6.37 ± 4.56 *	0.36 ± 0.31	0.43 ± 0.02	0.33 ± 0.13 *	0.57 ± 0.32
	Med	0.42	6.37	0.22	0.43	0.30	0.57
Cr	AM ± SD	0.03 ± 0.01	0.70 ± 0.52 *	0.04 ± 0.06	0.03 ± 0.01	0.03 ± 0.02 *	0.04 ± 0.02
	Med	0.03	0.70	0.02	0.03	0.03	0.04

**Table 3 pathogens-12-01039-t003:** Macroelement and microelement concentrations (mg/L) in the urine of immunosuppressed mice 8, 16, and 24 days post *Acanthamoeba* sp. infection (dpi) compared to the control group (AS, immunosuppressed *Acanthamoeba* sp. infected mice; CS, immunosuppressed uninfected control group mice; AM arithmetic mean; SD standard deviation; *p* level of significance; * *p* < 0.01, ** *p* < 0.02, *** *p* < 0.03 for the significance of the difference vs. control (Mann-Whitney U test).

Element	Parameter	AS			CS		
		8	16	24	8	16	24
Ca	AM ± SD	89.8 ± 71.5	60.5 ± 22.5	49.2 ± 46.2	50.9 ± 34.2	60.5 ± 22.5	56.6 ± 20.6
	Med	69.6	47.3	40.9	45.2	47.3	56.6
P	AM ± SD	31.0 ± 313.7	669.7 ± 826.0	3052.8 ± 6298.5	506.8 ± 88.4	626.9 ± 954.1	573.7 ± 526.7
	Med	199.1	386.5	160.7	506.8	217.9	573.7
Na	AM ± SD	1999.8 ± 729.8	2658.8 ± 709.2 **	2305.2 ± 1272.0	2619.5 ± 176.1	2145.2 ± 848.3 **	1696.5 ± 218.3
	Med	1829.4	2360.3	3168.3	2584.6	2060.6	1696.5
K	AM ± SD	8307.6 ± 2135.6	11,037.0 ± 1682.9	9371.8 ± 5627.5	9449.5 ± 1897.9	10,843.9 ± 1673.4	9592.8 ± 5898.4
	Med	8312.3	12,001.7	7263.4	9449.5	10,394.0	11,976.9
Mg	AM ± SD	400.4 ± 129.8	390.4 ± 135.2	373.0 ± 241.3	506.8 ± 88.4	426.0 ± 103.5	420.5 ± 115.8
	Med	403.9	428.2	282.4	506.8	395.7	420.5
Mn	AM ± SD	0.07 ± 0.04	0.05 ± 0.04	0.04 ± 0.03	0.03 ± 0.01	0.05 ± 0.04	0.07 ± 0.04
	Med	0.06	0.04	0.03	0.03	0.04	0.07
Zn	AM ± SD	8.25 ± 5.12 ***	3.74 ± 2.06	3.72 ± 2.74	2.19 ± 1.35 ***	4.92 ± 3.64	4.75 ± 1.54
	Med	8.98	3.35	2.91	1.89	3.73	4.75
Cu	AM ± SD	0.350 ± 0.154	0.27 ± 0.082	0.41 ± 0.43	0.20 ± 0.03	0.34 ± 0.11	0.35 ± 0.15
	Med	0.347	0.24	0.26	0.20	0.43	0.35
Fe	AM ± SD	1.67 ± 1.37 *	1.01 ± 0.480	2.22 ± 2.98	0.42 ± 0.22 *	1.00 ± 0.57	0.88 ± 0.46
	Med	1.35	0.78	0.98	0.37	0.88	0.88
Cr	AM ± SD	0.09 ± 0.05 ***	0.05 ± 0.02	0.05 ± 0.04	0.03 ± 0.03 ***	0.09 ± 0.04	0.06 ± 0.03
	Med	0.09	0.06	0.03	0.03	0.08	0.06

## Data Availability

The data presented in this study are available on request from the corresponding author.

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
