# Peer review of "Macroelement and Microelement Levels in the Urine in Experimental Acanthamoebiasis"

_pathogens, 2023, doi:10.3390/pathogens12081039_

Round 1

Reviewer 1 Report

This study looks at urine constituents and experimental Acanthamoeba infection. It is very well written and completely clear with the exception of Table 2 which is very difficult to read due to the tabulation but this is a production issue. The results justify the author's claims.

Author Response

Thank you for your review of our paper. We have answered each of your point below

This study looks at urine constituents and experimental Acanthamoeba infection. It is very well written and completely clear with the exception of Table 2 which is very difficult to read due to the tabulation but this is a production issue. The results justify the author's claims. Thank you we changed table 2.

Reviewer 2 Report

The purpose of this research was to evaluate the relationship between the levels of macroelements: calcium (Ca), phosphorus (P), sodium (Na), potassium (K), magnesium (Mg), and microelements such as: manganese (Mn), zinc (Zn), copper (Cu), iron (Fe), and chromium (Cr) in urine during experimental acanthamoebiasis, taking into account the immune status of the host.

It is a well written article with a good experimental design, good detection techniques for the micro and macroelements sought (using certified reference material to measure the quality of the analytical process), and good statistical analysis.

A strain of Acanthamoeba sp. It was used to cause acanthamebiasis in mice. Strain AM 22, isolated from the bronchial aspirate of a hemato-oncology patient suffering from acute septic shock (Łanocha et al., 2009). For this reason, authors are asked to use the term Acanthamoeba sp. instead Acanthamoeba spp. (the latter means several unknown species).

There are a mistake in table 3, results are about AS, (immunosuppressed Acanthamoeba sp. infected mice); CS, immunosuppressed uninfected control group mice. Authors repeated A and C of table 2. To make tables easy to read, it suggested eliminate the line with min-max results, so it will more easy find * p<0.01, **p< 0.02, ***p<0.03 for the significance of difference vs control.

Althoug, "there are studies that show that urinary trace element analysis can be used as an indicator of infection/disease severity [28-34], but the relationships between urinary trace element levels and parasitic infections, including acanthamoebiasis, remain relatively unknown". It is the first time that experimentally is shown that free-living amoebas can affect the excretion of macro and microelements in urine.

Because the experiments were done with an unknown species of Acanthamoeba, should be used Acanthamoeba sp., if they were done with several species, spp. In this case, the Acanthamoeba spp. must be changed by Acanthamoeba sp. throughout the article, as in the title of Table 2.

Usar upper letter for elements concentrations... in Table 1.

In Tble 3, change A for AS and C for CS.

Please consider eliminate the min-max line in both tables 2 and 3.

Author Response

Review 2

The corrections were marked with blue color throughout the article. 

Thank you for your review of our paper. We have answered each of your point below

The purpose of this research was to evaluate the relationship between the levels of macroelements: calcium (Ca), phosphorus (P), sodium (Na), potassium (K), magnesium (Mg), and microelements such as: manganese (Mn), zinc (Zn), copper (Cu), iron (Fe), and chromium (Cr) in urine during experimental acanthamoebiasis, taking into account the immune status of the host.

It is a well written article with a good experimental design, good detection techniques for the micro and macroelements sought (using certified reference material to measure the quality of the analytical process), and good statistical analysis. Thank you.

A strain of Acanthamoeba sp. It was used to cause acanthamebiasis in mice. Strain AM 22, isolated from the bronchial aspirate of a hemato-oncology patient suffering from acute septic shock (Łanocha et al., 2009). For this reason, authors are asked to use the term Acanthamoeba sp. instead Acanthamoeba spp. (the latter means several unknown species). We have changed the term according to reviewer's suggestions.

There are a mistake in table 3, results are about AS, (immunosuppressed Acanthamoeba sp. infected mice); CS, immunosuppressed uninfected control group mice. Authors repeated A and C of table 2. To make tables easy to read, it suggested eliminate the line with min-max results, so it will more easy find * p<0.01, **p< 0.02, ***p<0.03 for the significance of difference vs control. We have changed the Table legend according to reviewer's suggestions.

Although "there are studies that show that urinary trace element analysis can be used as an indicator of infection/disease severity [28-34], but the relationships between urinary trace element levels and parasitic infections, including acanthamoebiasis, remain relatively unknown". It is the first time that experimentally is shown that free-living amoebas can affect the excretion of macro and microelements in urine. We have changed the passage according to the reviewer's suggestions.

Because the experiments were done with an unknown species of Acanthamoeba, should be used Acanthamoeba sp., if they were done with several species, spp. In this case, the Acanthamoeba spp. must be changed by Acanthamoeba sp. throughout the article, as in the title of Table 2. We have changed the term according to reviewers suggestions.

Usar upper letter for elements concentrations... in Table 1. We changed it according to the reviewer's suggestion.

In Table 3, change A for AS and C for CS. We changed it according to reviewer's suggestion.

Please consider eliminate the min-max line in both tables 2 and 3. We have changed it according to reviewer's suggestion.

Reviewer 3 Report

Dear authors,

While I do not want to imply that this study is uninteresting, I think the research idea for this study has to be much better explained.  In stark contrast to the pathogens causing the diseases referred to in the Introduction, like sleeping sickness or schistosomiasis, Acanthamoeba is a free living microorganism. It may colonize various organs in case of accidental contact AND severe immunodeficiency, but infection of the kidneys has only very rarely been described. As said, it is conceivable that there are changes in the excretion of various elements in the course of infection, and it is, of course generally interesting to see which elements are found in higher concentrations and which ones in lower concentrations - if there is a certain pattern behind this. But the authors also need to explain, how can or could the amoebas affect the excretion of these elements and what can be concluded from the alterations observed.

Other issues:

has this ameba strain (AM 22) been deposited somewhere or is it accessible somewhere?

lines 167-169: what is meant here, if both, immunocompetent AND immunocompromised have the highest concentrations, who has the lowest or lower?

line 180: significant differences compared to what?

line 220: how was the grade of infection evaluated? 

lines 241-243: this is too vague, and the free living Acanthamoeba cannot be compared to parasites, which have obligatory life cycles involving different hosts.

line 245: for sure, concentrations of trace elements in the urine in many diseases follow a pattern, but why should they in Acanthamoeba infection? 

The Discussion in very interesting and nicely describes the importance of various macro and microelements in various diseases, but does not really describe the significance of the own results. How do all these studies relate to the current study?

line 351: really upregulation or rather higher concentration?

line 370: how? This is a very interesting aspect of this study, but this has to explained in more detail. How could this be done?

The English is generally fine, but there are some minor flaws here and there, e.g. missig articles, singular/plural mistakes etc.

Author Response

Review 3

The corrections were marked with red color throughout the article. 

Thank you for your review of our paper. We have answered each of your points below.

While I do not want to imply that this study is uninteresting, I think the research idea for this study has to be much better explained.  In stark contrast to the pathogens causing the diseases referred to in the Introduction, like sleeping sickness or schistosomiasis, Acanthamoeba is a free living microorganism. It may colonize various organs in case of accidental contact AND severe immunodeficiency, but infection of the kidneys has only very rarely been described. As said, it is conceivable that there are changes in the excretion of various elements in the course of infection, and it is, of course generally interesting to see which elements are found in higher concentrations and which ones in lower concentrations - if there is a certain pattern behind this. But the authors also need to explain, how can or could the amoebas affect the excretion of these elements and what can be concluded from the alterations observed.

According to our knowledge, the results of this multi-elemental analysis are the first regarding the excretion of trace elements in urine during acanthamoebiasis in both immunocompetent and immunocompromised hosts. The findings of our research are significant for the knowledge in the area and may have potential applications in the future. However, interpreting the results is challenging due to the lack of comparative data available in the scientific literature. Nevertheless, these obtained results undoubtedly expand our understanding of element excretion in urine during amoebic infection and may serve as a basis for further research on element metabolism.

Other issues:

has this ameba strain (AM 22) been deposited somewhere or is it accessible somewhere?

Sequence data  from strain AM 22 was deposited in GenBank and is available under the following reference numer: strain AM22, GQ342607.

lines 167-169: what is meant here, if both, immunocompetent AND immunocompromised have the highest concentrations, who has the lowest or lower?

We have clarified this passage according to the reviewer's suggestion.

line 180: significant differences compared to what?

We have clarified it in the text.

line 220: how was the grade of infection evaluated? 

Infected animals were monitored daily for clinical signs, behavior, appetite, and mortality. The following elements were assessed: activity of the mice, feeding behavior, appearance of the fur, hunched position, ataxia, and tremors. A scoring system was used to evaluate the clinical status, with points ranging from 0 to 2. The average score from three measures was calculated. Mice with an average total score of 3 were classified as severely sick (reaching the humane endpoint), those with scores between 4 and 7 were considered moderately sick, and mice with a total score of over 8 were deemed healthy (without symptoms).

To confirm the mice's infection with amoebas, kidneys were collected from euthanized animals, and the amoebas were isolated and analyzed. The results of these studies have been published in our previous publication (Łanocha-Arendarczyk et al., 2018; Kot et al., 2021).

  1. Łanocha-Arendarczyk N, Baranowska-Bosiacka I, Kot K, Gutowska I, Kolasa-Wołosiuk A, Chlubek D, Kosik-Bogacka D. Expression and Activity of COX-1 and COX-2 in Acanthamoeba sp.-Infected Lungs According to the Host Immunological Status. Int J Mol Sci. 2018 Jan 2;19(1):121. doi: 10.3390/ijms19010121.
  2. Kot K., Kosik-Bogacka D., Łanocha-Arendarczyk N., Ptak M., Roszkowska P., Kram A. Histological changes, in the kidneys and heart in experimental Acanthamoebiasis in immunocompetent and immunosuppressed hosts, Folia Biologica-Krakow, vol. 69, no. 4, 2021, pp. 167-178.

lines 241-243: this is too vague, and the free living Acanthamoeba cannot be compared to parasites, which have obligatory life cycles involving different hosts. We delate this statments.

line 245: for sure, concentrations of trace elements in the urine in many diseases follow a pattern, but why should they in Acanthamoeba infection? 

Disseminated acanthamoebiasis is characterized by the presence of multi-organ and multi-symptom lesions, which progress rapidly and often result in the death of the host. Our previous animal studies have provided clear evidence that amoebae, carried through the bloodstream, infiltrate the kidneys and cause their dysfunctions (Kot et al., 2020; Kot et al., 2021). As the accuracy of creatinine as a marker for kidney damage in parasitic diseases is limited, there is a need for new and more effective biomarkers.          The analysis of trace elements in urine has been utilized to assess the body's condition during inflammatory processes. Therefore, we aimed to investigate whether infection with Acanthamoeba affects the concentrations of macro and microelements in the urine of hosts with different immunological statuses.

We have added this information in Discussion.

  1. Kot, K.; Kosik-Bogacka, D.; Wojtkowiak-Giera, A.; Kolasa-Wołosiuk, A.; Łanocha-Arendarczyk, N. The expression of TLR2 and TLR4 in the kidneys and heart of mice infected with Acanthamoeba spp. Parasit. Vectors. 2020, 13, 480. 407
  2. Kot, K.; Kupnicka, P.; Witulska, O.; Czepan, A.; Łanocha-Arendarczyk, N.A.; Łanocha, A.A.; Kosik-Bogacka, D.I. Potential biomarkers in diagnosis of renal acanthamoebiasis. Int. J. Mol. Sci. 2021, 22, 6583.

The Discussion in very interesting and nicely describes the importance of various macro and microelements in various diseases, but does not really describe the significance of the own results. How do all these studies relate to the current study?

We changed according to reviewers suggestions.

line 351: really upregulation or rather higher concentration?

We changed according to reviewers suggestions.

line 370: how? This is a very interesting aspect of this study, but this has to explained in more detail. How could this be done?

Certainly, in the future, it would be worth attempting to use specific trace element values as early biomarkers of Acanthamoeba infection. However, this work has certain limitations. Our research constitutes a preliminary study conducted on an animal model. Due to the high mortality rate of animals and the need for a humanitarian endpoint, we had a limited number of mice available for experimentation. Conducting a multifactorial analysis of trace elements, renal profile, and hormonal levels with a larger sample size could help confirm the hypothesis that changes in urinary element concentrations in Acanthamoeba sp. infected hosts may be useful for monitoring the host's prognosis. We have added some relevant information in the manuscript.

Round 2

Reviewer 3 Report

All of my minor issues have been addressed satisfyingly. But I still do not really understand the hypothesis behind this study, at least not with the arguments given. Why should there be a specific pattern in the concentration of trace elements in the urine in severly immunocompromised patients who have opportunistic amebas growing in the brain or even the entire body (as in the mouse model); namely a pattern other than in any other severe infection and/or other than in immunocompromised individuals in general.

But as said, of course, it advances the knowledge to know which elements go up or down in concentration during the course of infection. While I do not think that this will help diagnostics in any way (as certainly not specific enough, and also, lab diagnostics based on qPCR is already highly sensitive and specific). But possibly this knowledge may advance treatment. It would have been interesting to discuss this in more detail.

Author Response

Thank you for another review and comments that have contributed to a more in-depth analysis of our research and more cautious drawing of conclusions.

All of my minor issues have been addressed satisfyingly. But I still do not really understand the hypothesis behind this study, at least not with the arguments given. Why should there be a specific pattern in the concentration of trace elements in the urine in severly immunocompromised patients who have opportunistic amebas growing in the brain or even the entire body (as in the mouse model); namely a pattern other than in any other severe infection and/or other than in immunocompromised individuals in general.

Following the reviewer's comments, we have added a paragraph in the introduction regarding the significance and necessity of research in hosts with different immunological statuses. Additionally, we have removed the paragraph about the kidney and its biomarkers, as it could be confusing for the manuscript's readers.

But as said, of course, it advances the knowledge to know which elements go up or down in concentration during the course of infection. While I do not think that this will help diagnostics in any way (as certainly not specific enough, and also, lab diagnostics based on qPCR is already highly sensitive and specific). But possibly this knowledge may advance treatment. It would have been interesting to discuss this in more detail.

In the conclusions, we have removed speculations about the utilization of our results in the diagnostic process of Acanthamoeba infection. Instead, we have added a paragraph discussing the potential application of the knowledge about the host's mineral status in treatment.